# Zingerone-Induced Autophagy Suppresses IL-1β Production by Increasing the Intracellular Killing of *Aggregatibacter actinomycetemcomitans* in THP-1 Macrophages

**DOI:** 10.3390/biomedicines11082130

**Published:** 2023-07-28

**Authors:** Yuri Song, Jin Chung

**Affiliations:** 1Department of Oral Microbiology, School of Dentistry, Pusan National University, Yangsan-si 50612, Republic of Korea; luckcute@pusan.ac.kr; 2Oral Genomics Research Center, Pusan National University, Yangsan-si 50612, Republic of Korea

**Keywords:** *Aggregatibacter actinomycetemcomitans*, autophagy, interleukin (IL)-1β, periodontitis, zingerone

## Abstract

Periodontitis is caused by the inflammation of tooth-supporting tissue by pathogens such as *Aggregatibacter actinomycetemcomitans.* Interleukin-1β (IL-1β), a pro-inflammatory cytokine, triggers a series of inflammatory reactions and promotes bone resorption. The aim of this study was to examine the molecular mechanism and anti-inflammatory function of zingerone, a dietary phenolic found in *Zingiber officinale,* on periodontal inflammation induced by *A. actinomycetemcomitans.* Zingerone attenuated *A. actinomycetemcomitans*-induced nitric oxide (NO) production by inhibiting the expression of inducible nitric oxide synthase (iNOS) in THP-1 macrophages. Zingerone also inhibited the expression of tumor necrosis factor (TNF)-α, IL-1β, and their signal pathway molecules including the toll-like receptor (TLR)/mitogen-activated protein kinase (MAPKase). In particular, zingerone suppressed the expression of absent in melanoma 2 (AIM2) inflammasome components on IL-1β production. Moreover, zingerone enhanced autophagosome formation and the expressions of autophagy-associated molecules. Interestingly, zingerone reduced the intracellular survival of *A. actinomycetemcomitans*. This was blocked by an autophagy inhibitor, which reversed the decrease in IL-1β production by zingerone. Finally, zingerone alleviated alveolar bone absorption in an *A. actnomycetemcomitans*-induced periodontitis mice model. Our data suggested that zingerone has potential use as a treatment for periodontal inflammation induced by *A. actinomycetemcomitans.*

## 1. Introduction

Periodontitis is a chronic inflammatory disease induced by microbial imbalance between symbiotic commensals and pathogens [1]. *Aggregatibacter actinomycetemcomitans (A. actinomycetemcomitans)*, a facultative anaerobic gram-negative rod, is a keystone pathogen present at elevated higher levels in the subgingival plaque of periodontitis patients [2]. *A. actinomycetemcomitans* expresses various virulence factors such as lipopolysaccharide, leukotoxin, collagenase, and IgG protease, which are linked to the pathogenesis of periodontal disease [3,4,5]. Furthermore, the invasion of epithelial cells by *A. actinomycetemcomitans* can protect bacterial cells from mechanical removal, antibiotics, immune cell phagocytosis, and antibody binding [6]. After invasion, *A. actinomycetemcomitans* induces several pathogenic mechanisms in human leukocytes [3]. *A. actinomycetemcomitans* degrades immunoproteins, secrete inflammatory proteins, and kills immune cells, which may all contribute to its survival in infected hosts [6].

Neutrophils and monocyte/macrophages play important roles in immune homeostasis by regulating inflammation [7]. Macrophages express a variety of inflammatory cytokines, including TNF-α, IL-1β, and IL-6 in periodontitis-affected sites [8]. A clinical study has reported significantly higher levels of IL-1β and TNF-α in the gingival crevicular fluid of individuals with periodontitis and peri-implantitis compared to both healthy subjects and those with healthy implants [9]. Of these, IL-1β is a potent signaling molecule that extends the inflammatory cascade and causes tissue damage [8]. IL-1β secretion is regulated by a two-step mechanism: the transcription of the pro-IL-1β gene through the MAPKase/TLRs pathway and the activation of inflammasomes, which convert pro-IL-1β to IL-1β [10]. NO, produced by iNOS, is a critical mediator of inflammatory response [11] and its serum concentrations are elevated in periodontitis [12]. Cytokines produced by inflammatory mediators are beneficial in appropriate amounts, but excessive production causes the development of periodontitis [13].

Autophagy is a natural, regulated cell mechanism that removes unnecessary or dysfunctional components [14]. Many studies described the beneficial effects of autophagy in vitro in the context of defense against invading pathogens such as *Streptococcus, Shigella flexneri, Mycobacterium tuberculosis,* and *Salmonella typhimurium* [15]. In a previous study, we reported that *A. actinomycetemcomitans* induces autophagy in THP-1 macrophages and that *A. actinomycetemcomitans*-induced autophagy down-regulates inflammatory response [16]. Furthermore, the induction of autophagy inhibits *M. tuberculosis* survival within infected macrophages. Thus, it appears several intracellular bacteria have evolved ways of avoiding autophagy [17,18], and that the induction of autophagy provides a means of controlling inflammatory response. 

As regards the treatment of periodontitis, traditional non-surgical methods are performed as the initial approach for periodontitis patients [19]. Non-surgical methods include scaling and root planning, chemical and antibiotic agents, and photostimulation (laser/LED treatment) [19]. Several studies have reported antibiotic adjuncts such as amoxicillin and metronidazole are effective treatments for *A. actinomycetemcomitans* [20]. Ideally, antibiotics should be specific for periodontal pathogens, non-allergic, and nontoxic; however, most antibiotics have side effects [21]. In particular, artificial antimicrobial agents induce antimicrobial resistance and destroy commensal bacteria [21]. Thus, natural herbal medicines with anti-oxidant or anti-inflammatory properties are attractive substitutes [22]. Zingerone [4-(4-hydroxy-3-methoxyphenyl) bu-tan-2-one] is an active component found in dry ginger rhizomes [23], and *Zingiber officinale* is widely used as a dietary spice that has potent anti-inflammatory, anti-oxidative, and anticancer properties [24]. Furthermore, it has been reported that ginger has antibacterial activity against anaerobic Gram-negative bacteria such as *Porphyromonas gingivalis, Porphyromonas endodontalis, and Prevotella intermedia* [25]. However, no report has been issued on the anti-inflammatory effect and the precise molecular mechanisms of zingerone on periodontal inflammation. 

Thus, the aim of this study was to investigate the effect of zingerone on *A. actinomycetemcomitans*-induced inflammatory responses in vitro and in vivo. We set a null hypothesis as: “The administration of zingerone does not have any effect on *A. actinomycetemcomitans*-induced inflammatory response, both in vitro in THP-1 macrophages and in vivo using a mice model of periodontitis”. To test the hypothesis, we compared zingerone-treated groups with control groups by statistically assessing various parameters; the expression of inflammatory cytokines, activation of signaling pathways, modulation of autophagy, intra-cellular survival of *A. actinomycetemcomitans*, and alveolar bone absorption.

## 2. Materials and Methods

### 2.1. Bacterial Culture

*Aggregatibacter actinomycetemcomitans* (ATCC 33384) was grown in tryptic soy (TBS) broth (Becton Dickinson, Franklin Lakes, NJ, USA) with 1% yeast (LPS solution, Seoul, Republic of Korea) at 37 °C in a 5% CO_2_ incubator. An OD of 0.25 (650 nm) was determined to correlate to 1 × 10^9^ CFU/mL. *A. actinomycetemcomitans* was harvested by centrifugation at 5000 rpm for 5 min, resuspended in RPMI media (Thermo Fisher Scientific, Waltham, MA, USA), and used to infect the macrophages at a multiplicity of infection (MOI) of 50.

### 2.2. Cell Culture

THP-1 cells, a human monocyte cell line, were cultured in RPMI media (Thermo Fisher Scientific, Waltham, MA, USA) supplemented with 10% fetal bovine serum and differentiated into macrophages using a 50 nM Phorbol 12-myristate 13-acetate (Sigma-Aldrich, St. Louis, MO, USA) overnight. Zingerone (purity ≥ 98%) was purchased from Sigma-Aldrich. THP-1 macrophages were pretreated with zingerone for 30 min, and then infected with *A. actinomycetemcomitans*. 3-Methyladenine (3MA; Sigma-Aldrich) was used to block autophagy induction. THP-1 macrophages were pretreated with 3MA for 30 min and then infected with *A. actinomycetemcomitans*.

### 2.3. MTT Assay

Cell viability was determined using an MTT (3-(4,5-dimethylthiazol-2-yl)-2,5-diphenyl tetrazolium bromide) assay kit (Sigma-Aldrich). Briefly, 5 × 10^4^ cells were seeded in 96-well plates overnight and then treated with various concentrations of zingerone. After incubation for 24 h, the medium was replaced with a serum-free medium containing 500 μg/mL MTT reagents for 2 h. The conversion of MTT reagent into chromogenic formazan was determined using a microplate reader at 570 nm. Absorbances of treated cells were calculated as percentages of solvents of zingerone.

### 2.4. Quantitative Real-Time Polymerase Chain Reaction (PCR)

Total RNA was extracted with the RNeasy Mini kit (Qiagen, Valencia, CA, USA). cDNA was synthesized using a reverse transcription system (Bioneer Co., Daejeon, Republic of Korea) and 1 μg of total RNA. cDNA was amplified in a total volume of 20 μL containing 2× PCR master mix (Applied Biosystems, Waltham, MA, USA) using gene-specific primers on an ABI 7500 real-time PCR system (Applied Biosystems). Relative mRNA levels were calculated using the 2^-ΔΔCt^ method. Duplicate measurements were made on ≥3 each sample per group. The data were normalized relative to those for GAPDH (glyceraldehyde-3-phosphate dehydrogenase) as an endogenous control. The primer sequences (Bioneer Co.) used were: iN-OS(F), 5′-TGTGCTCTTTGCCTGTATGC-3′; iNOS(R), 5′-TTGCCAAAC-GTACTGGTCAC-3′; TNF-α(F), 5′-ACGGCATGGATCTCAAAGAC-3′; TNF-α(R), 5′-TGAGATAGCAA ATCGGCTGAC-3′; IL-1β(F), 5′-GAGTGTGGATCCCAAGCAAT-3′; IL-1β(R), 5′-CTTGTGCTCTGCTTGTGAGG-3′; AIM2(F), 5′-TCAGCTGAAATGAGTCCTGC-3′; AIM2(R), 5′-CTTGGGTCTCAAACGTGAAGG-3′; ASC(F), 5′-TACCTGGAGACCTACGGCG-3′; ASC(R), 5′-TATAAAGTGCAGGCCCTGGTG-3′; Caspase-1(F), 5′-ATCCGTTCCATGGGTGAAGG-3′; Caspase-1(R), 5′-CGTGCTGTCAGAGGTCTTGT-3′; GAPDH(F), 5′-ACCCAGAAGACTGTGGATGG-3′; and GAPDH(R), 5′-ACACATTGGGGGTAGGAACA-3′.

### 2.5. Cytokine Assay

The levels of IL-1β and TNF-α were measured using a human IL-1β ELISA MAX Deluxe kit (BioLegend, San Diego, CA, USA), and human TNF-α ELISA MAX Deluxe kit, respectively. All ELISAs were performed following the manufacturer’s instruction.

### 2.6. Immunoblot Analysis

Cells were harvested and lysed in RIPA buffer (Cell Signaling Technology, Danvers, MA, USA) containing protease inhibitor cocktail (Sigma-Aldrich). Proteins samples were separated using 10–15% SDS-PAGE and then transferred to membranes (Millipore, Burlington, MA, USA), which were then probed with primary and secondary antibodies. Membranes were developed using a chemiluminescence solution (GE Healthcare, Chicago, IL, USA) in a LAS-4000 Lumino-imaging unit (Fujifilm, Tokyo, Japan). Immunoblot band intensities were quantified using NIH ImageJ software (Version 1.53t, Fujifilm), and results were presented as intensity ratios versus β-actin. The antibodies used were as follows; anti-β-actin (Santa Cruz Biotechnology, Dallas, TX, USA), anti-human iNOS (Santa Cruz Biotechnology), anti-human TLR2 (Santa Cruz Biotechnology), anti-human TLR4 (Santa Cruz Biotechnology), NF-κB (Cell Signaling Technology), MAPK Family antibody sampler kit (Cell Signaling Technology), anti-human AIM2 (Cell Signaling Technology), anti-human ASC (Cell Signaling Technology), pro-Caspase-1 (Cell Signaling Technology), anti-human IL-1β (R&D Systems, Minneapolis, MN, USA), Beclin1 (Santa Cruz Biotechnology), ATG5 (Santa Cruz Biotechnology), and LC3 (Cell Signaling Technology).

### 2.7. Confocal Laser Scanning Microscopy for ASC Speck Observation

THP-1/ASC-GFP cells were seeded in 8-well chambers (Thermo Fisher Scientific) and challenged with *A. actinomycetemcomitans* with or without zingerone. ASC forms a single 1- to 2-mm supramolecular assembly in macrophages in response to proinflammatory stimuli [26]. In unstimulated cells, ASC-GFP was evenly distributed in cytoplasm and nuclei. After stimulation, ASC-GFP fluorescence was observed by confocal laser scanning microscopy (Carl Zeiss, Oberkochen, Germany) as large bright spots in the cytoplasm, indicating the formation of ASC speck. The percentages of cells exhibiting ASC speck were calculated with respect to total cells.

### 2.8. Nitric Oxide Measurement

NO production was determined by measuring the amount of nitrite in culture supernatants using Griess reagent (Sigma-Aldrich). Briefly, equal volumes of cell supernatants and Griess reagent were mixed to a 96-well plate at room temperature for 10 min. Absorbances were measured at 570 nm using a microplate reader (Tecan, Männedorf, Switzerland) and NO concentrations were determined using a sodium nitrite-derived standard curve.

### 2.9. Growth Curve

*A. actinomycetemcomitans* was cultured in TBS at 37 °C in an atmosphere containing 5% CO2 up to the late log phase of growth (optical density [OD], 1). *A. actinomycetemcomitans* was added to 1 × 10^8^ CFU per well in a 96-well plate in 200 μL of medium containing various concentration of zingerone. The growth curve was plotted by measuring absorbances at 660 nm using a microplate reader (Tecan). ODs of each well were calculated against the TBS medium.

### 2.10. Viable Cell Count (VCC)

To determine bacterial adhesion, THP-1 macrophages were pretreated with zingerone (40 μM) for 4 h, infected with *A. actinomycetemcomitans* (MOI 50) for 90 min, and washed with warm PBS to remove non-invasional bacteria, and lysed in sterile distilled water for 20 min. Cell lysates were diluted with TSB medium, and then spread in agar plates. After incubation for 24 h, the number of viable cells was counted.

To determine the bacterial killing after *A. actinomycetemcomitans* infection, THP-1 macrophages were infected with *A. actinomycetemcomitans* (MOI 50) for 90 min and then treated with antibiotics for 30 min to eliminate non-invasive bacteria. Invaded macrophages were subsequently incubated with or without zingerone for 4 h. Number of the viable cells was counted.

To assess the autophagy effect of zingerone, THP-1 macrophages were infected with *A. actinomycetemcomitans* (MOI 50) for 90 min, and treated with antibiotics for 30 min to eliminate non-invasive bacteria. The invaded macrophages were subsequently incubated with 3MA for 30 min to block autophagy induction, and then treated with zingerone for 4 h, and the number of viable cells was counted.

### 2.11. Autophagy Determination by Confocal Laser Scanning Microscopy

Cells were stained with fluorescent dyes using the CYTO-ID Autophagy Detection Kit (Enzo Life Sciences, Farmingdale, NY, USA), which contains a cationic amphiphilic dye that, without transfection, precisely stains autophagic vacuoles (pre-autophagosomes, autophagosomes, and autolysosomes) green with minimal lysosome staining. Whole macrophages were stained with Hoechst-33342 (a nuclear stain) to determine total cell populations. Briefly, the cells were incubated with the CYTO-ID Detection agent at 37 °C for 15 min on a light block and then observed by confocal laser scanning microscopy. Percentages of autophagic cells were calculated with respect to total cell numbers.

### 2.12. Animal Study

C57BL/6 mice (6 weeks) were purchased from Korea Animal Technology (Koatech; Pyeongtaek, Republic of Korea) and maintained at 23 ± 2 °C under a relative humidity of 60 ± 5% and a 12 h light–dark cycle. During the week before the experiment, the mice were given sulfamethoxazole-trimethoprim (Ratiopharm, Ulm, Germany) ad libitum in drinking water for 3 days to remove resident oral flora, and then rested for 4 days. For animal experiments, mice were divided into four groups (*n* = 6–7 per group): (1) a control group (orally administered 4% carboxymethyl cellulose (Sigma-Aldrich) (sham); (2) an *A. actinomycetemcomitans*-infected group (inoculated with 100 µL of *A. actinomycetemcomitans* (5 × 10^8^ colony-forming units/mL) suspended in 4% carboxymethyl cellulose; (3) an *A. actinomycetemcomitans* + zingerone (400 mg/kg) group; and (4) a zingerone (400 mg/kg) group.

To generate the periodontitis model, mice were inoculated three times a week for two weeks by oral gavage. To confirm the effect of zingerone, mice were treated with zingerone (400 mg/kg) three times weekly for four weeks by oral gavage. The mice were euthanized with CO_2_ gas at week 13. All animal procedures were performed in accordance with the protocols approved by the Institutional Animal Care and Use Committee of Pusan National University (PNU-2022-0108).

### 2.13. Micro-CT Scanning and Assessment of Alveolar Bone

Mouse mandibular bones were prepared to observe alveolar bone resorption. Three-dimensional images were scanned by micro-CT (InspeXio SMX-90CT; Shimadzu Science, Tokyo, Japan) in slices at 90 kV and 110 μA using a 0.5 mm aluminum attenuation filter). Scans were reconstructed to generate three-dimensional models of region of interest (ROI). ROIs included alveolar bone body from the most mesial to the most distal aspects of the lower three molar roots. Three-dimensional images were generated in a standardized position using TRI/3D Bone software (version 7.0, Ratoc, Tokyo, Japan) and defined anatomical references. Alveolar bone loss was determined by summing the area from the cementoenamel junction (CEJ) to the alveolar bone crest (ABC) of each tooth using ImageJ. A single blinded examiner performed all measurements.

### 2.14. Statistics

All data are presented as means ± the standard deviation (SD). GraphPad Prism 9.5 software (GraphPad, San Diego, CA, USA) was used for statistical analysis. Normality of the variables was checked via the Shapiro–Wilk test, and then performed by Student’s *t*-test for normally distributed samples or Mann–Whitney test for nonparametric samples. The comparisons among multiple groups were analyzed by one-way analysis of variance (ANOVA) with a Dunnett post-hoc test. Statistical significance was considered at *p* < 0.05, indicating significant differences between groups.

## 3. Results

### 3.1. Zingerone Suppressed A. actinomycetemcomitans–Induced NO Production by Inhibiting iNOS Expression in THP-1 Macrophage

To assess the cytotoxic effect of zingerone, THP-1 macrophages were treated with various concentrations (10–80 μM) for 24 h. Zingerone did not induce any cytotoxicity at any concentration (*p* = 0.377) (Figure 1A). NO metabolite levels were increased (*p* < 0.001) by *A. actinomycetemcomitans* infection, but these increases were significantly and dose-dependently suppressed (*p* < 0.05) by zingerone (Figure 1B). Furthermore, *A. actinomycetemcomitans* induced the mRNA (*p* < 0.001) and protein expression of iNOS, whereas zingerone dose-dependently suppressed this induction (*p* < 0.05) (Figure 1C,D). These results suggested that zingerone attenuated the *A. actinomycetemcomitans*-induced production of NO by suppressing iNOS expression in THP-1 macrophages.

### 3.2. Zingerone Suppressed A. actinomycetemcomitans–Induced TLR/MAPKase Activation

Next, we investigated the mRNA and secreted levels of the proinflammatory cytokines IL-1β and TNF-α on *A. actinomycetemcomitans*-infected THP-1 macrophages. After infection, the RNA level of IL-1β (*p* = 0.004) and TNF-α (*p* = 0.001) increased compared to controls, and these were significantly and dose-dependently inhibited (*p* < 0.05) by zingerone (Figure 2A,C). Moreover, IL-1β (*p* = 0.014) and TNF-α (*p* = 0.001) cytokines were significantly upregulated by the infection, which was reduced (*p* < 0.05) by zingerone (Figure 2B,D).

To determine the signal pathway of the anti-inflammatory effects of zingerone on *A. actinomycetemcomitans*-infected THP-1 macrophages, we examined the expressions of sensing molecules. *A. actinomycetemcomitans* significantly increased TLR2 and TLR4 expression compared to the uninfected control cells, but zingerone decreased the expression of these molecules (Figure 2E).

Next, we examined the effect of zingerone on the expressions of downstream signal molecules, such as nuclear factor (NF)-κB and MAPKase which includes the extracellular signal-regulated kinases (ERK), p38, and c-Jun N-terminal kinases (JNK). *A. actinomycetemcomitans* significantly increased the levels of phosphorylated JNK, p38, ERK, and NF-κB compared to uninfected control cells. Zingerone significantly suppressed these phosphorylated JNK, p38, and ERK, but not that of NF-κB (Figure 2F). These results indicated that zingerone suppressed *A. actinomycetemcomitans*–induced TLR/MAPKase activation.

### 3.3. Zingerone Suppressed A. actinomycetemcomitans-Induced AIM2 Inflammasome, Leading to Reduced IL-1β Production

To examine the effect of zingerone on inflammasome components, which are critical for IL-1β production induced by *A. actinomycetemcomitans* infections, we examined the expression of the inflammasome complex. The AIM2, ASC, caspase-1, and IL-1β mRNA and protein levels were upregulated (*p* < 0.05) by *A. actinomycetemcomitans* infection, but zingerone suppressed (*p* < 0.05) these increases (Figure 3A,B). Furthermore, the numbers of ASC speck cells were increased (*p* = 0.034) by *A. actinomycetemcomitans* infection, but these were also decreased (*p* = 0.022) by zingerone (Figure 3C). In addition, *A. actinomycetemcomitans* increased intracellular pro-IL-1β expression and IL-1β secretion in the culture supernatants, and zingerone also suppressed these increases (Figure 3D). These results suggested that zingerone inhibited AIM2 inflammasome activation, which led to the suppression of IL-1β production in *A. actinomycetemcomitans*-infected THP-1 macrophages.

### 3.4. Zingerone Enhanced Intracellular Bacterial Killing by Inducing Autophagy in A. actinomycetemcomitans-Infected THP-1 Macrophages

To investigate whether the action mechanism of zingerone affected bacterial survival, we counted the number of viable bacteria. Zingerone at all concentrations did not influence (*p* = 0.056) the growth of *A. actinomycetemcomitans* for up to 24 h (Figure 4A). Meanwhile, zingerone significantly inhibited (*p* = 0.003) the intracellular survival of *A. actinomycetemcomitans* within THP-1 macrophages without bacterial adhesion to THP-1 macrophages was not affected (*p* = 0.990) (Figure 4B). These results indicated that zingerone enhanced the intracellular killing of *A. actinomycetemcomitans*-infected THP-1 macrophages.

To identify the intracellular killing mechanism, we investigated the effect of zingerone on autophagy. *A. actinomycetemcomitans* infection induced (*p* < 0.001) autophagosome formation, which was significantly enhanced (*p* = 0.017) by zingerone (Figure 4C). Zingerone also increased autophagy-associated proteins, including Beclin1, ATG5/12, and LC3-II compared to *A. actinomycetemcomitans*-infected cells (Figure 4D).

To confirm that the enforced autophagic effect by zingerone was related to reduced IL-1β expression, the autophagy inhibitor was used. 3MA treatment more increased (*p* < 0.001) intracellular bacteria survival in *A. actinomycetemcomitans* infected cells, and blocked (*p* = 0.002) the inhibitory effect of zingerone on intracellular bacteria survival (Figure 4E). Moreover, 3MA treatment attenuated (*p* < 0.001) the zingerone-induced reductions in IL-1β production (Figure 4F). Overall, these results suggested that zingerone decreased IL-1β production by enhancing intracellular bacterial killing in THP-1 macrophages by augmenting autophagy.

### 3.5. Zingerone Administration Ameliorated Alveolar Bone Resorption in A. actinomycetemcomitans-Infected Periodontitis Mice Model

To examine the effect of zingerone on *A. actinomycetemcomitans*-induced periodontitis, alveolar bone loss in mice was measured using micro-CT. The mice were treated with zingerone three times weekly for four weeks and then inoculated with *A. actinomycetemcomitans* (Figure 5A). Alveolar bone resorption was greater (*p* < 0.001) in *A. actinomycetemcomitans*-infected mice than in uninfected control mice, but zingerone administration inhibited (*p* = 0.002) this bone resorption (Figure 5B). This result suggested that zingerone inhibited alveolar bone absorption in *A. actinomycetemcomitans* infected periodontitis mice.

## 4. Discussion

Periodontitis is the result of disruption of the homeostasis of host-pathogen interactions [27]. A key component of a dental biofilm is the presence of the keystone pathogen such as *Porphyromonas gingivalis, Filifactor alocis*, and *Aggregatibacter actinomycetemcomitans* [28]. Furthermore, in peri-implantitis biofilms, a higher prevalence of *Aggregatibacter actinomycetemcomitans* and *Prevotella intermedia* was detected compared with healthy implants [29]. Bacteria are necessary for the onset of disease, and then it ultimately can damage the periodontal tissues is the host’s inflammatory response to microbial attacks [30].

In susceptible individuals, when the host response fails to resolve an early infection in a timely manner, cytokine overproduction destroys tissue [31,32]. Therefore, controlling inflammatory cytokines is important for preventing and treating periodontitis.

First, IL-1β and TNF-α are elevated in periodontitis patients, and thus are considered representative inflammatory cytokines in this background [33]. NO is also a critical mediator of inflammation [34], and is synthesized from L-arginine by NOS [34]. Of the three NOS isoforms (nNOS, eNOS, iNOS), iNOS is an important regulator of the immune response and produces large amounts of NO in response to proinflammatory stimuli [35]. NO destroys invading microorganisms, but its excessive expression may be cytotoxic to host tissues and cause tissue breakdown [36]. In the present study, the anti-inflammatory effect of zingerone in *A. actinomycetemcomitans*-infected THP-1 macrophages reduced the *A. actinomycetemcomitans*-induced productions of NO, TNF-α, and IL-1β (Figure 1 and Figure 2). Similarly, gingerol, the major pungent compound in the rhizomes of ginger, inhibited the production of TNF-α, IL-1β, and IL-12 in LPS-stimulated macrophages [37]. Thus, it seems that the suppressive effect of zingerone on NO, TNF-α, and IL-1β production indicates it might have a beneficial effect on *A. actinomycetemcomitans*-induced periodontal inflammation.

Macrophages recognize microorganisms by sensing pathogen-associated molecular patterns (PAMPs) through pattern recognition receptors (PRRs) such as TLRs [38]. These receptors activate the MAPKase and NF-κB signaling pathways, which lead to the production of pro-inflammatory mediators [39]. In a previous study, we reported that *A. actinomycetemcomitans* infection activated the TLR2, TLR4, and MAPKase signaling pathways [16]. Thus, we examined whether zingerone influenced the TLR signaling pathway in a background of *A. actinomycetemcomitans*-induced inflammation. Zingerone inhibited *A. actinomycetemcomitans*-induced expressions of TLR2 and TLR4 (Figure 2E) and attenuated the *A. actinomycetemcomitans*-induced phosphorylations of all three MAPKases (JNK, p38, and ERK), but did that of NF-κB (Figure 2F). Another study reported that zingerone exhibited protective effects against LPS-induced liver damage by inhibiting the TLR4-mediated inflammatory pathway [40]. These results suggest that zingerone might have an anti-inflammatory effect by suppressing TLR/MAKase signaling.

IL-1β production is closely related to the pathogenesis of periodontitis [41]. IL-1β causes tissue destruction and bone resorption by stimulating the proliferation and multi-nucleation of osteoclasts [42]. Thus, IL-1β levels can be used as a diagnostic marker of periodontitis. IL-1β release is strictly regulated by two signal pathways [43]. The first pathway requires MAKase and NF-κB activation via TLR sensing [44] and results in the production of pro-IL-1β, whereas the second pathway involves the activation of inflammasomes, which convert pro-IL-1β to IL-1β [44]. Inflammasomes are composed of NLR or AIM2 family sensors, ASC, and procaspase-1. Upon activation, NLR or AIM2 family members bind to the adaptor protein ASC, which then recruits procaspase-1 for activation. Activated caspase-1 cleaves the proform of IL-1β to its mature and secreted forms [45]. Previously, we reported that *A. actinomycetemcomitans* activated AIM2 inflammasome [44,46]. Based on this result, in the current study, we investigate that zingerone inhibited *A. actinomycetemcomitans*-induced AIM2 inflammasome activation by reducing the expressions of AIM2, ASC, and caspase-1 at the mRNA and protein levels (Figure 3). Zingerone also reduced ASC speck formation and, finally the secreted form of lL-1β. These results suggest that the zingerone suppressed the IL-1β production by inhibiting AIM2 inflammasome activation in *A. actinomycetemcomitans*-infected THP-1 macrophages.

To elucidate the action mechanism of the anti-inflammatory effect of zingerone, the association between zingerone and autophagy was examined. Zingerone significantly enhanced the number of autophagic cells and the expressions of Beclin1, ATG5, and LC3 induced by *A. actinomycetemcomitans* infection (Figure 4C,D). Interestingly, zingerone increased the intracellular *A. actinomycetemcomitans* killing activity within THP-1 macrophages, and this was reduced by 3MA, an autophagy inhibitor (Figure 4B,E). In line with this result, decreased IL-1β production by zingerone was prevented by 3MA (Figure 4F). Taken together, these results suggest that zingerone enhanced autophagy and that this increased the intracellular killing of *A. actinomycetemcomitans*, which eventually reduced IL-1β secretion. This is the first study to report that the molecular mechanism responsible for the anti-inflammatory activity of zingerone involves enhanced intracellular killing of bacteria by autophagy in macrophages.

Finally, the effect of zingerone on alveolar bone resorption resulting from periodontal inflammation was tested in periodontitis mice model. We investigated the alveolar bone loss by micro-CT after an *A. actinomycetemcomitans* infection with or without a zingerone administration. As was expected, zingerone reduced *A. actinomycetemcomitans*-induced alveolar bone resorption, which concurs with a previous report that zingerone inhibited LPS-induced systemic inflammation in NF-κB transgenic mice [47]. The present study demonstrated that zingerone inhibits periodontal inflammation induced by *A. actinomycetemcomitans* infection in C57BL/6 mice.

The limitation of this study is that it only confirmed the anti-inflammatory effect of zingerone on a single periodontal pathogen. Periodontitis is caused by a biofilm composed of various bacteria. Therefore, to achieve clinical applicability, further research is needed to investigate the anti-inflammatory effect of zingerone on other pathogens involved in periodontitis.

## 5. Conclusions

In summary, we revealed for the first time the anti-inflammatory potential of zingerone and its molecular signaling pathway. This study shows that zingerone reduced *A. actinomycetemcomitans*-induced IL-1β production by suppressing AIM2 inflammasome activation. Moreover, zingerone increased intracellular bacterial killing by inducing autophagy. Eventually, zingerone mitigated IL-1β production induced by *A. actinomycetemcomitans* through autophagy-mediated intracellular bacterial killing in THP-1 macrophages. Thus, this study presents a novel molecular mechanism for the control of periodontal inflammation by zingerone and suggests zingerone is a potential treatment for *A. actinomycetemcomitans*-related periodontitis. 

## Figures and Tables

**Figure 1 biomedicines-11-02130-f001:**
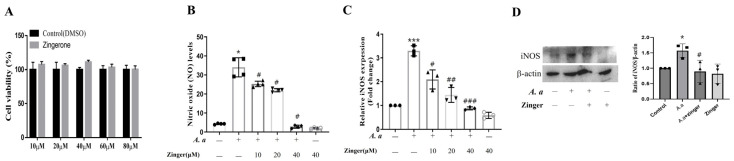
Zingerone suppressed *A. actinomycetemcomitans*–induced NO production by inhibiting iNOS expression in THP-1 macrophages. (**A**) The effect of zingerone on cell viability was determined using an MTT assay. Results are presented as means ± SD (*n* = 6). One-way ANOVA was used for analysis. (**B**,**C**) THP-1 macrophages were pretreated with indicated concentration of zingerone for 30 min and then infected with *A. actinomycetemcomitans* (MOI 50) for 24 h. Results are presented as means ± SD (**B**) NO levels in culture media were measured using a Griess assay (*n* = 4). (**C**) mRNA levels of iNOS were analyzed by real-time PCR (*n* = 3). Relative mRNA expression levels were normalized to those of GAPDH. (**D**) THP-1 macrophages were pretreated with zingerone (40 μM) for 30 min, and then infected with the *A. actinomycetemcomitans* (MOI 50) for 24 h. Protein levels of iNOS were analyzed using a Western blot. Densitometry of the band intensity was performed. Results are representative of three independent experiments. * *p* < 0.05, *** *p* < 0.001 control versus *A. actinomycetemcomitans* infection, # *p* < 0.05, ## *p* < 0.01, ### *p* < 0.001 *A. actinomycetemcomitans* infection versus *A. actinomycetemcomitans* with zingerone.

**Figure 2 biomedicines-11-02130-f002:**
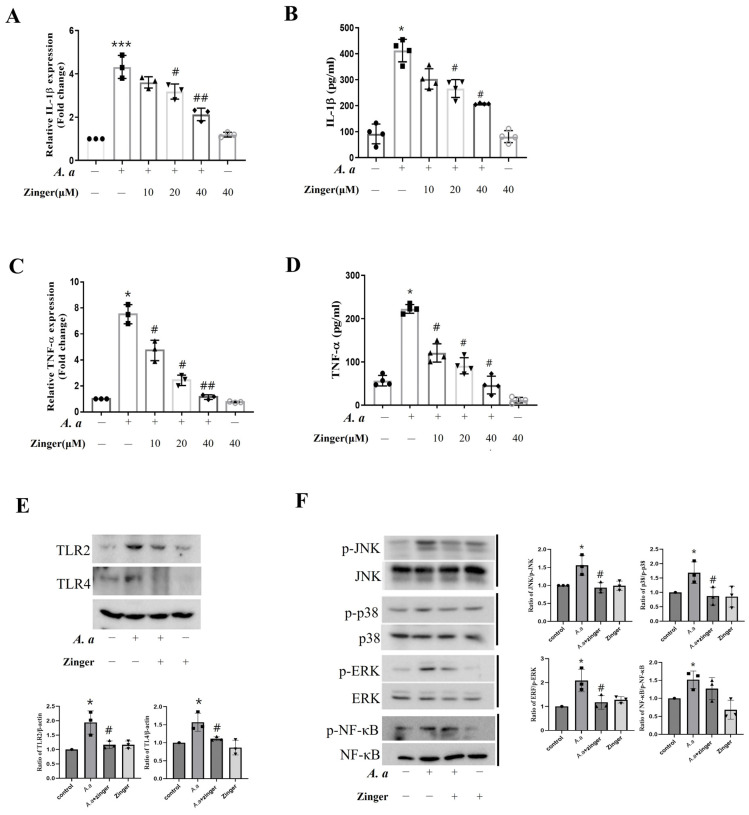
Zingerone suppressed *A. actinomycetemcomitans*–induced TLR/MAPKase activation. (**A**–**D**) THP-1 macrophages were pretreated with the indicated concentration of zingerone for 30 min and then infected with the *A. actinomycetemcomitans* (MOI 50) for 24 h. (**A**,**C**) mRNA levels of IL-1β were analyzed by real-time PCR (*n* = 3). Relative mRNA expression levels were normalized versus GAPDH. (**B**,**D**) Secreted levels of IL-1β in the culture medium were measured by ELISA (*n* = 3). (**E**) Protein levels of TLR2 and TLR4 protein expression were detected by Western blot assay. Densitometry of the band intensity was performed. (**F**) THP-1 macrophages were pretreated with the zingerone (40 μM) for 30 min, and then infected with the *A. actinomycetemcomitans* (MOI 50) for 30 min. The NF-κB and MAPKs were detected by Western blot assay. Densitometry of the band intensity was performed. Results are representative of three individual experiments. Results are presented as means ± SD (*n* = 3). * *p* < 0.05, *** *p* < 0.001 control versus *A. actinomycetemcomitans* infection, # *p* < 0.05, ## *p* < 0.01 *A. actinomycetemcomitans* infection versus *A. actinomycetemcomitans* with zingerone.

**Figure 3 biomedicines-11-02130-f003:**
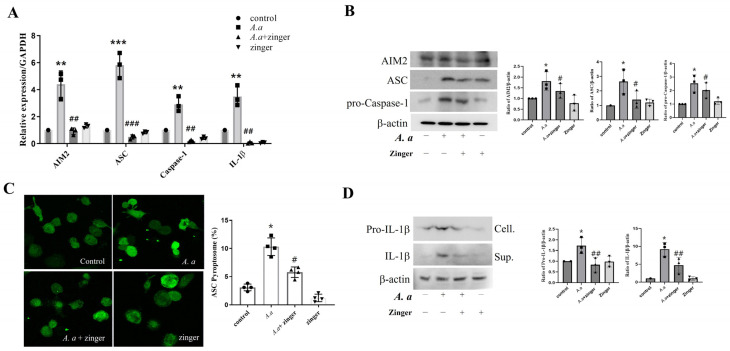
Zingerone suppressed *A. actinomycetemcomitans*-induced AIM2 inflammasome, which inhibits IL-1β production. THP1 macrophages were pretreated with zingerone (40 μM) for 30 min, and then infected with the *A. actinomycetemcomitans* (MOI 50) for 24 h. (**A**) mRNA levels of AIM2, ASC, Caspase-1, and IL-1β were analyzed by real-time PCR (*n* = 3). Relative mRNA expression levels were normalized to those of GAPDH. (**B**) Protein levels of AIM2, ASC, and Caspase−1 protein expression were detected by Western blot assay. Densitometry of the band intensity was performed. (**C**) ASC speck was observed and photographed under a fluorescence confocal microscope. Confocal images (400× magnification). The graph shows the percentages of cells containing ASC speck (**D**) Protein levels of IL-1β in cell lysate and supernatant of secretion by Western blot assay. Densitometry of the band intensity was performed. Results are representative of three individual experiments. Results are presented as means ± SD (*n* = 3). * *p* < 0.05, ** *p* < 0.01, *** *p* < 0.001 control versus *A. actinomycetemcomitans* infection, # *p* < 0.05, ## *p* < 0.01, ### *p* < 0.001 *A. actinomycetemcomitans* infection versus *A. actinomycetemcomitans* with zingerone.

**Figure 4 biomedicines-11-02130-f004:**
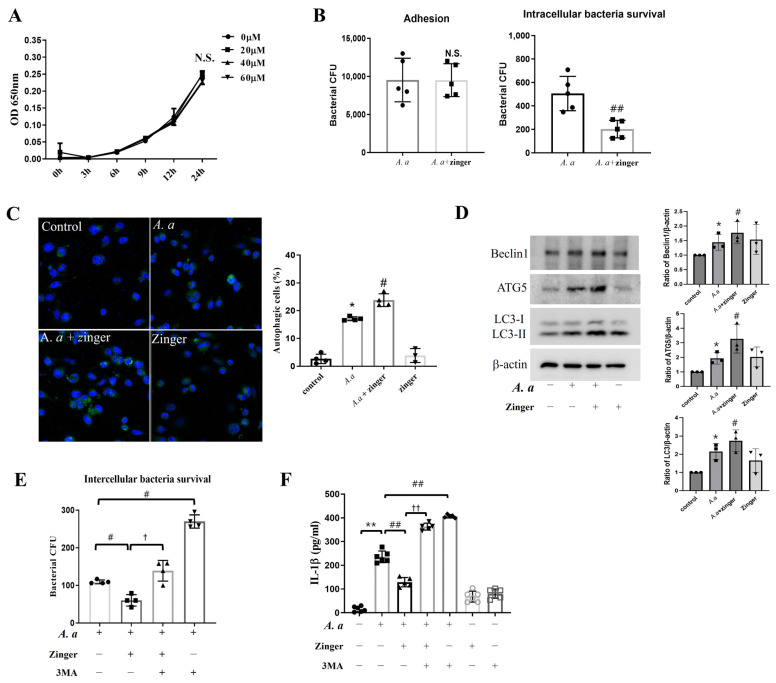
Zingerone enhanced intracellular bacterial killing by inducing autophagy in *A. actinomycetemcomitans*-infected THP-1 macrophages. (**A**) The growth curve of *A. actinomycetemcomitans* was measured at OD 650 nm during incubation with zingerone (20, 40, 60 μM). Results are presented as means ± SD (*n* = 6). One-way ANOVA was used for analysis. (**B**) Bacteria adhesion (**left**) and intracellular bacteria survival (**right**) were counted using VCC as described in Materials and Methods (*n* = 5). (**C**,**D**) THP-1 macrophages were pretreated with zingerone (40 μM) for 30 min, and then infected with *A. actinomycetemcomitans* (MOI 50) for 24 h. (**C**) Autophagic cells were stained by CYTO-ID and analyzed using confocal microscopy. Confocal images (400× magnification). Mean CYTO-ID fluorescence intensities are shown by the bar graphs. (**D**) Protein levels of Beclin1, ATG5/12, and LC3-II expression were detected by Western blot. Densitometry of the band intensity was performed. (**E**) Intracellular bacteria survival was counted using VCC (*n* = 4). (**F**) THP-1 macrophages were pretreated with 3MA (5 mM) and zingerone (40 μM) for 30 min, and then infected with *A. actinomycetemcomitans* (MOI 50) for 24 h. Secreted levels of IL-1β in culture medium were measured by ELISA (*n* = 3). Results are representative of three individual experiments. Results are presented as means ± SD. N.S. = Not Significant, * *p* < 0.05, ** *p* < 0.01 control versus *A. actinomycetemcomitans* infection, # *p* < 0.05, ## *p* < 0.01 *A. actinomycetemcomitans* infection versus *A. actinomycetemcomitans* with zingerone or 3MA, † *p* < 0.05, †† *p* < 0.01 *A. actinomycetemcomitans* infection with zingerone versus *A. actinomycetemcomitans* infection with 3MA+zingerone.

**Figure 5 biomedicines-11-02130-f005:**
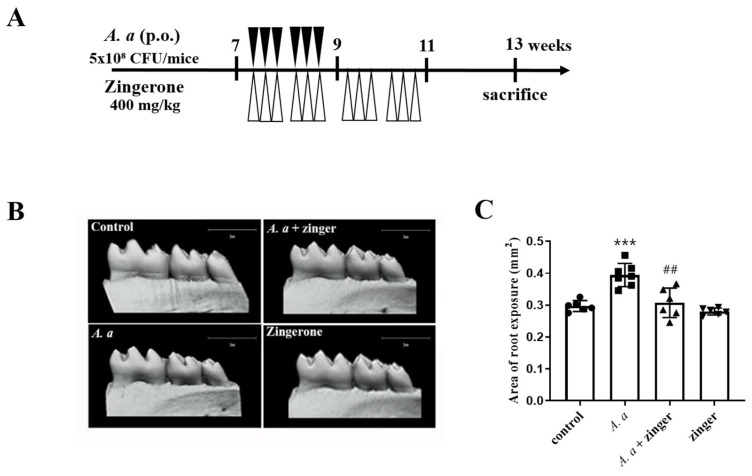
Zingerone administration attenuated the alveolar bone resorption in the *A. actinomycetemcomitans*-induced periodontitis mice models. (**A**) Experimental design for periodontitis mice models (**B**) Micro-CT image of alveolar bone in mice. Representative images of mandibles in the uninfected control group, the *A. actinomycetemcomitans*-infected group, the *A. actinomycetemcomitans* infection with zingerone group, and the zingerone group. (**C**) Areas of bone loss were measured, and the mean values of each group were represented as a graph. Results are presented as means ± SD (*n* = 6–7). *** *p* < 0.001 control versus *A. actinomycetemcomitans* infection, ## *p* < 0.01 *A. actinomycetemcomitans* infection versus *A. actinomycetemcomitans* with zingerone.

## Data Availability

Data for this study, though not available in a public repository, will be made available upon reasonable request.

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
