# Peer review of "Zingerone-Induced Autophagy Suppresses IL-1β Production by Increasing the Intracellular Killing of Aggregatibacter actinomycetemcomitans in THP-1 Macrophages"

_biomedicines, 2023, doi:10.3390/biomedicines11082130_

Round 1
Reviewer 1 Report (New Reviewer)
Dear Authors,
you made a great work!

Minor revisions are suggested.
Author Response
The paper is an in vitro study on the Zingerone-induced autophagy suppresses IL-1β production by increasing the intracellular killing of Aggregatibacter actinomycetemcomitans in THP-1 macrophages.
The Authors made a great work in terms of methodology and the paper sounds scientific and well written.
However, some improvements are mandatory before acceptance.
The abstract is well written, complete and summary in its various aspects. The keywords are complete and appropriate.
Please check double spaces and punctuation.
Response: Thank you for your comment. We have checked again.
In the introduction:
-
the introduction is well written and comprehensive, I think it could be improved by making it more specific to the study at hand.
-
“Macrophages express a variety of inflammatory cytokines, including TNF- , IL-1 , and IL-6 in periodontitis-affected sites [9].” I think it is interesting to underline how this inflammatory aspect is extremely interesting to be considered, particularly in the light of recent research on this topic conducted above all at the implant level, where the pro-inflammatory levels are higher, and also considering the interleukins which more than others determine peri-implant and periodontal resorption, as underlined by: “Guarnieri R, Reda R, Zanza A, Miccoli G, Nardo DD, Testarelli L. Can Peri-Implant Marginal Bone Loss Progression and a-MMP-8 Be Considered Indicators of the Subsequent Onset of Peri-Implantitis? A 5-Year Study. Diagnostics (Basel). 2022 Oct 26;12(11):2599. doi: 10.3390/diagnostics12112599.” and I think it is also important to consider the direction of the latest research trends regarding the main microorganisms involved in the progression of this pathology: “Mahendra J, Mahendra L, Mugri MH, Sayed ME, Bhandi S, Alshahrani RT, Balaji TM, Varadarajan S, Tanneeru S, P ANR, Srinivasan S, Reda R, Testarelli L, Patil S. Role of Periodontal Bacteria, Viruses, and Placental mir155 in Chronic Periodontitis and Preeclampsia-A Genetic Microbiological Study. Curr Issues Mol Biol. 2021 Jul 29;43(2):831-844.”
Response: Thank you for your comment. We have added a description of the cytokines aspect of peri-implantitis and periodontitis (lines 44-47) as well as the latest research on the microorganisms in the progression of this pathology (lines 391-396). We believe that the revised version provides a better and more accurate background for our study.
Materials and methods are clear and well explained. Different aspects are analyzed with a dedicated statistical test. The authors did a great job in the explication of all the variables identified and included in the study.
Results are easy to understand and comprehensive. All the studied characteristics were reported in tables which are clear and concise. The tables are complete and understandable.
Discussion: this section is complete and evaluates the outcome of different papers present in literature. The overall is comprehensive, concise and complete in its various aspects.
Conclusions are concise and clear.
Bibliography is formatted respecting the journal’s requirements and no improper citations are evidenced.
Figures and labels are clear and easy to comprehend.
English is clear and easy to understand.

Reviewer 2 Report (New Reviewer)
This is an interesting study on a novel antibacterial agent against A.a., however, several issues must be addressed. Please see the enclosed PDF

Author Response
Response to Reviewer 2 Comments
1. The authors should specify other treatment methods besides antibiotics, such as laser, photoactiviation, I suggest: Mocanu, R.C; Martu, M.A
Profiles of Patients with Dental Prosthetic Treatment and Periodontitis before and after Phostoactivation therpay- randomized clinical trial microorganism
Response: Thank you for your comment. We have added other treatment methods of periodontitis in introduction (line 65-68).
2. The authors should state the null hypothesis
Response: Thank you for your comment. We have added the null hypothesis in introduction (line 85-89)
3. The images are too small, the authors should separate them (at most 3 by 3) and enlarge
Response: Thank you for your comment. We have enlarged the Figure 1.
4. Please enlarge this image
Response: Thank you for your comment. We have been enlarged the Figure 3C.
5. Enlarge image
Response: Thank you for your comment. We have been enlarged the Figure 4C.
6. In the discussion section the authors do not compare their results to those from the existing literature. This section must be improved upon.
Response: Thank you for your comment. We have additionally described the comparison of our results with those of previous studies (line 423-425). Besides, the comparison results are mentioned in line 409 and 459.

Reviewer 3 Report (New Reviewer)
The manuscript entitled "Zingerone-induced autophagy suppresses the IL-1β production by increasing intracellular killing of Aggregatibacter actinomycetemcomitans in THP-1 macrophages" brings information about efficacy of Zingerone in Aggregatibacter actinomycetemcomitans (Aa) induced in periodontitis mice model.
I suggest The title of the manuscript to contain "periodontitis induced in mice" since this is an animal experimental study. By this detail in the tile, the work is easier find by the scientist who wants to follow the method described the authors for other researches.
Abstract
Line 5 - induced production of NO by inhibiting iNOS? Please reformulate this sentence because is to long and confusing.
Please specify at the end of the Introduction the aim of the study, because you only wrote a description of methods and results, that are not necessary in this chapter of your manuscript.
Material and Methods
Please specify the methods for animal euthanasia (line 210-211).
Please write at the end of the manuscript study limitations, and what is needed more for translational research to the clinical studies.
Author Response
Respons e to Reviewer 3 Comments
1. The manuscript entitled "Zingerone-induced autophagy suppresses the IL-1β production by increasing intracellular killing of Aggregatibacter actinomycetemcomitans in THP-1 macrophages" brings information about efficacy of Zingerone in Aggregatibacter actinomycetemcomitans (Aa) induced in periodontitis mice model.
I suggest The title of the manuscript to contain "periodontitis induced in mice" since this is an animal experimental study. By this detail in the tile, the work is easier find by the scientist who wants to follow the method described the authors for other researches.
Response: Thank you for your comment. Since the effect of autophagy was not examined in the mouse model, the title of result 5 was changed “Zingerone administration ameliorated alveolar bone resorption in A. actinomycetemcomitans-infected periodontitis mice model. (line 371) ”instead of the title of the manuscript.
Abstract
2. Line 5 - induced production of NO by inhibiting iNOS? Please reformulate this sentence because is to long and confusing.
Response: Thank you for your comment. We revised this sentence (line 15).
3. Please specify at the end of the Introduction the aim of the study, because you only wrote a description of methods and results, that are not necessary in this chapter of your manuscript.
Response: Thank you for your comment. We corrected them according to your comment (line 81-89).
Material and Methods
4. Please specify the methods for animal euthanasia (line 210-211).
Response: Thank you for your comment. We have added the method for animal euthanasia (line 216).
5. Please write at the end of the manuscript study limitations, and what is needed more for translational research to the clinical studies.
Response: Thank you for your comment. We have additionally described the study limitations and the necessary points for clinical research (line 464-468).

Reviewer 4 Report (New Reviewer)
This study shows that zingerone attenuates the inflammatory response by enhancing intracellular killing of A. actinomycetemcomitans in macrophages. The study includes an animal study that was conducted according to the rules. Thus, this study presents a new molecular mechanism for the control of periodontal inflammation by zingerone and suggests that zingerone is a potential treatment for A. actinomycetemcomitans-associated periodontitis. I would like to see the results of the action of this drug on other microorganisms, especially on the pathogen of the oral cavity - Candida. The conclusion is very short. Please, increase it.
Author Response
Response to Reviewer 4 Comments
This study shows that zingerone attenuates the inflammatory response by enhancing intracellular killing of A. actinomycetemcomitans in macrophages. The study includes an animal study that was conducted according to the rules. Thus, this study presents a new molecular mechanism for the control of periodontal inflammation by zingerone and suggests that zingerone is a potential treatment for A. actinomycetemcomitans-associated periodontitis. 1) I would like to see the results of the action of this drug on other microorganisms, especially on the pathogen of the oral cavity - Candida. 2) The conclusion is very short. Please, increase it.
Response: Thank you for your comment. 1) We agree that it is very important. For clinical use, it is necessary to confirm the effect of zingerone on another pathogens including Candida albicans. We describe it in the end of Discussion. Further research on this topic would be of great interest. 2) And, we have additionally described the conclusion (471-477).

Round 2
Reviewer 2 Report (New Reviewer)
The manuscript has been improved
Reviewer 4 Report (New Reviewer)
Accept in present form. Authors answered all questions.
This manuscript is a resubmission of an earlier submission. The following is a list of the peer review reports and author responses from that submission.
Round 1
Reviewer 1 Report
In this paper, several assays have been carried out in order to analyze if zingerone can be used in the treatment of periodontal inflammation induced by A. actinomycetemcomitans. The paper is well structured and written however, for its publication, several modifications should be carried out:
Introduction
- Introduction could be expanded a little more, including what studies there are currently, what they have studied, how they have done it and what results they have obtained.
Materials and methods
- This part is clear; however, authors could include the purity of reagents
- Line 124. Write “A. actinomycetemcomitans” in italics
Results and discussion
- These sections are well written, structured and results are robust.
Conclusions
- Where is this part of the paper?
Therefore, the reviewer suggests that this paper could be included in the consideration for publication in Biomedicines with minor modifications.
Author Response
[Please see the attachment; manuscript & response to reviewer1 comment]
Response to Reviewer 1 Comments
Comments and Suggestions for Authors
In this paper, several assays have been carried out in order to analyze if zingerone can be used in the treatment of periodontal inflammation induced by A. actinomycetemcomitans. The paper is well structured and written however, for its publication, several modifications should be carried out:
Response: Thank you for your valuable comments. As recommended, the manuscript was revised.
-
Introduction
- Introduction could be expanded a little more, including what studies there are currently, what they have studied, how they have done it and what results they have obtained.
Response 1: Thank you for your comment. As recommended, introduction was revised (line 79-86).
-
Materials and methods
- This part is clear; however, authors could include the purity of reagents
Response 2: Thank you for your comment. As recommended, the purity was written down as “Zingerone (purity ³ 98%) was purchased from Sigma-Aldrich.” (line 100).
- Line 124. Write “A. actinomycetemcomitans” in italics
Response 3: Thank you. It is corrected as recommended.
-
Results and discussion
- These sections are well written, structured and results are robust.
Response 4: Thank you for your comment.
-
Conclusions
- Where is this part of the paper?
Response 5: Thank you for your comment. As recommended, the conclusion was described at the end of the manuscript (lines 450-455).

Reviewer 2 Report
This study investigated the potential of zingerone to suppress the inflammation caused by Aggregatibacter actinomycetemcomitans (Aa)during periodontitis. Administration of zingerone attenuated the inflammatory response to Aa by inhibiting the production of iNOS, TNF and IL-1B. Administration of zingerone also appeared to attenuate bone resorption in mice infected with Aa. There are some reasonable results in there but they are not presented very well.
I have some concerns with the presentation of the paper. There are numerous grammatical errors making it difficult to understand what the author is trying to say. The Introduction is too short and doesn’t mention the previous work done at Pusan National University that this study obviously builds on. There is no mention of the intracellular nature of the bacterial species.
There are lots of details missing from the methods. For example, RT-PCR results are presented in the results but there are no details in the methods. The list of antibodies used in the Western blots is incomplete. There is no mention of how many bacterial cells are combined with the various concentrations of zingerone used in the growth curves. It is unclear how many times the experiments were conducted. Details are missing from the description of ASC speck observation and autophagy determination. For example, 3-MA was included as an inhibitor in the autophagy assay but is not mentioned in the methods section.
There are also numerous flaws with the presentation of the results. For example, on line 258 there is a reference to Figure S1 but it is virtually identical to Figure 4D. The references to 4D and 4E don’t match the figure legend. 4D in the text is clearly 4E and 4E in the text is clearly 4F. Many of the images of the Western blots are hard to interpret and I am not convinced I would draw the same conclusions as the author. For example, I am not convinced that zingerone is reducing the amount of TLR2 as shown in Figure 2E. I also find it difficult to detect a difference in phosphorylated p38 in Figure 2F. This may be due to the quality of the image. The methods mention band intensities being quantified and presented as ratio relative to the intensity of B-actin (lines 93-94). To me this implies there should be numbers. Presentation of images of Western blots is semi-quantitative. More detail is needed in the legend for Figure 5 and the legend on the y axis of Figure 5B (should be 5C) should be more informative than mm2.
The discussion contains information that should be included in the introduction but is otherwise fine.
The quality of the English needs to be improved. There are numerous grammatical errors and a number of sentences did not make sense.
Author Response
"Please see the attachment"
Response to Reviewer 2 Comments
This study investigated the potential of zingerone to suppress the inflammation caused by Aggregatibacter actinomycetemcomitans (Aa)during periodontitis. Administration of zingerone attenuated the inflammatory response to Aa by inhibiting the production of iNOS, TNF and IL-1B. Administration of zingerone also appeared to attenuate bone resorption in mice infected with Aa. There are some reasonable results in there but they are not presented very well.
Point 1. I have some concerns with the presentation of the paper. 1) There are numerous grammatical errors making it difficult to understand what the author is trying to say. 2) The Introduction is too short and doesn’t mention the previous work done at Pusan National University that this study obviously builds on. 3) There is no mention of the intracellular nature of the bacterial species.
Response 1: Thank you for your comment. 1) First, I re-edited the English language, and then attached a confirmation certification in back page. As recommended, in the introduction section, 2) the previous work done at our lab was described (lines 58-61) and 3) the intracellular nature of the bacterial species was also described (lines 38-43). I believe that the revised introduction would provide better and more accurate experimental evidence for the study.
Point 2. There are lots of details missing from the methods. For example, 1) RT-PCR results are presented in the results but there are no details in the methods. 2) The list of antibodies used in the Western blots is incomplete. 3) There is no mention of how many bacterial cells are combined with the various concentrations of zingerone used in the growth curves. 4) It is unclear how many times the experiments were conducted. Details are missing from the description of 5) ASC speck observation and 6) autophagy determination. For example, 7) 3-MA was included as an inhibitor in the autophagy assay but is not mentioned in the methods section.
Response 2: Thank you for your comment. the methods as 1) RT-PCR (line 115-132), 2) the lists of antibodies of Western blots (lines 145-153), and 3) bacteria cells used in the growth curves (line 172) were described in Materials and Methods.
4) I accurately described the number of experiments in each figure legend as “Results are presented as means±SDs (n=6).” or “Results are representative of three individual experiments.”
I added more details about the methods for 5) ASC speck observation (lines 154-162), 6) autophagy determination (lines 194-202), and 7) 3-MA treatment in autophagy assay (lines 189-193).
Point 3. There are also numerous flaws with the presentation of the results. For example, 1) on line 258 there is a reference to Figure S1 but it is virtually identical to Figure 4D. The references to 4D and 4E don’t match the figure legend. 4D in the text is clearly 4E and 4E in the text is clearly 4F. 2) Many of the images of the Western blots are hard to interpret and I am not convinced I would draw the same conclusions as the author. For example, I am not convinced that zingerone is reducing the amount of TLR2 as shown in Figure 2E. I also find it difficult to detect a difference in phosphorylated p38 in Figure 2F. This may be due to the quality of the image. The methods mention band intensities being quantified and presented as ratios relative to the intensity of B-actin (lines 93-94). To me, this implies there should be numbers. The presentation of images of Western blots is semi-quantitative. 3) More detail is needed in the legend for Figure 5 and the legend on the y axis of Figure 5B (should be 5C) should be more informative than mm2.
Response 3: Thank you for your comment. 1) First of all, I would like to apologize for the mistake in the manuscript. The numbers of figures 4 were correctly revised.
2) The intensities for each band image of Western blot were added in the figures. The related original data is attached on page 16-24.
3) In figure 5B, “mm2” has been changed to “Area of root exposure (mm2)”
Your feedback helped me identify that information on several things was incorrect, and I greatly appreciate it. They were corrected in the revised version.
Point 4. The discussion contains information that should be included in the introduction but is otherwise fine.
Response 4: Thank you. I agree with the reviewer’s comment. The paragraph (‘autophagy’ content) in discussion transferred to the introduction.

Reviewer 3 Report
Yuri Song investigates the effect of Zingerone - a compound derived from ginger rhizomes - on THP-1 cell line-derived macrophages in the presence of A. actinomycetemcomitans (a pathogen associated with peroidontal disease). The manuscript was not very easy to read in some parts and there is a lot of information missing including methods, replicates, quantifications,... The statistical approach is not acceptable at this point.
Methods are missing in the materials and methods section (e.g. MTT assay, RT-qPCR - primers???). Also, there is no information on when the cells were harvested for the analysis of specific genes or proteins.
Statistics: How many animals were used for the animal experiments? In all other experiments n = 3. With such small numbers it is not appropriate to perform parametric statistical tests such as the student t-test or ANOVAs. In addition, individual data points should be shown on the graphs.
Western blot images should be quantified and replicates should be shown (attachement). The loading control is missing (e.g. actin) in figuere 2F and no molecular weight markers are shown in any of the western blots.
The language should be checked carefully. For example, the first lines of the abstract are not correct.
Author Response
"Please see the attachment."
Response to Reviewer 3 Comments
Yuri Song investigates the effect of Zingerone - a compound derived from ginger rhizomes - on THP-1 cell line-derived macrophages in the presence of A. actinomycetemcomitans (a pathogen associated with peroidontal disease). Point 1: The manuscript was not very easy to read in some parts and there is a lot of information missing including methods, replicates, quantifications,... The statistical approach is not acceptable at this point.
Response 1: Thank you for your comments. First, I re-edited the English language, and then attached a confirmation certification in back page. As recommended, the manuscript was revised.
Point 2: Methods are missing in the materials and methods section (e.g. 1) MTT assay, 2) RT-qPCR - primers???). Also, 3) there is no information on when the cells were harvested for the analysis of specific genes or proteins.
Response 2: Thank you for your comment. The methods as 1) MTT assay (lines 106-113), 2) RT-PCR (lines 115-132), and 3) cell harvest time in figure legends were added (e.g. 24 h).
Your feedback helped me identify that information on several things was incorrect, and I greatly appreciate it. They were corrected in the revised version.
Point 3: Statistics: 1) How many animals were used for the animal experiments? 2) In all other experiments n = 3. With such small numbers it is not appropriate to perform parametric statistical tests such as the student t-test or ANOVAs. 3) In addition, individual data points should be shown on the graphs.
Response 3: Thank you for your comment. 1) The animal number was described in the method as “The mice were divided into four groups (n= 6 per group) (line 208).”
2) And, I accurately described for our experimental statistics in the method as; “All data are presented as means ± the standard deviations (SD). Statistical compari-sons between the two groups were analyzed with unpaired Student’s t-test. Comparisons among multiple groups were analyzed with one-way ANOVA using GraphPad Prism software (GraphPad Software, San Diego, CA, USA). Statistical significance was consid-ered at the P-values <0.05, indicating significant differences between groups.”
Furthermore, I described the number of experiments at each figure legends as “The data are represented as mean ± SD (n=6 or 3).” and “The results represent those from one of three individual experiments.” Generally, in most experimental papers, three individual experiments with n=3 were performed, and present results from one of three individual experiments. In line with this, I present the reference papers published in biomedicine [1,2] and our previous paper [3, 4].
Reference
[1] Zhng, Y.; Fu, J.; Han, Yang. Et al. wo-Pore-Domain Potassium Channel TREK–1 Mediates Pulmonary Fibrosis through Macrophage M2 Polarization and by Direct Promotion of Fibroblast Differentiation. Biomedicines 2023, 11, 1279.
[2] Seo, J.; Lee, D. E.; Kim, S. M.; et al. Licochalcone A Exerts Anti-Cancer Activity by Inhibiting STAT3 in SKOV3 Human Ovarian Cancer Cells. Biomedicines 2023, 11, 1264.
[3] Lee, H.A.; Park, M. H.; Song, Y. et al. Role of Aggregatibacter actinomycetemcomitans-induced
autophagy in inflammatory response. J. periodontal. 2020, 91, 1682-1693.
[4] Lee, H.A.; Song, Y.; Park, M. H. et al. Catechin ameliorates Porphyromonas gingivalis-induced
inflammation via the regulation of TLR2/4 and inflammasome signaling. J. periodontal. 2020, 91, 661-670.
3) As recommended, I changed all graph to scatter plot graph that can represent individual data points.
Point 4: 1) Western blot images should be quantified and 2) replicates should be shown (attachement). 3) The loading control is missing (e.g. actin) in figuere 2F and no molecular weight markers are shown in any of the western blots.
Response 4: Thank you for your comment. 1) Figures were updated by adding the intensities of each band image, 2) I attached western blot images that I have repeatedly performed on page . And I believe that the revised version would provide better and more accurate experimental evidence for the study.
3) In figure 2F, changes in protein expression were analyzed by comparing the phosphorylation amount with the total amount. In line with this opinion, I present the reference papers [5, 6] on biomedicine that does not use actin. Nevertheless, I agree with your comment that checking actin together is an accurate experiment, so I will do that for further studies.
Reference
[5] Koga, A.; Thongsiri, C.; Kudo, D. et al. Mechanisms Underlying the Suppression of IL-1β Expression by Magnesium Hydroxide Nanoparticles. Biomedicines 2023, 11, 1291.
[6] Jeong, D.; Ko, W. K.; Kim, S. J.; et al. Lobeglitazone Exerts Anti-Inflammatory Effect in Lipopolysaccharide-Induced Bone-Marrow Derived Macrophage. Biomedicines 2021, 9, 1432.

Round 2
Reviewer 3 Report
Many of my major concerns remain and some additional ones have risen when seeing this revised version of the manuscript, specifically:
1. Western blot images are not reliable for several reasons:
a. The images of individual experiments should be quantified (including with actin staining as an internal control) and subsequently these results should be expressed as the ratio target/ actin. (in case of protein/ phosphoprotein – this ratio is also possible)
b. When looking at the repetition experiments, do not see the effect being repeated in the other blots/ experiments. This is why quantification is important.
2. Statistics:
The use of parametric statistics (descriptive: means, SD; tests: t-test and ANOVA) is only appropriate in case the data has a normal distribution. The data should first be tested for normal distribution and if found not-normal, non-parametric statistics should be used.
3. The fact that iNOS is an enzyme does not interfere with western-blot analysis. iNOS is a protein and all enzymes denature during SDS-PAGE, thereby loosing their catalytic activity.
The english has improved